# USCO-Solver: Solving Undetermined Stochastic Combinatorial Optimization Problems

**Guangmo Tong**

Department of Computer and Information Sciences

University of Delaware

`amotong@udel.edu`

## Abstract

Real-world decision-making systems are often subject to uncertainties that have to be resolved through observational data. Therefore, we are frequently confronted with combinatorial optimization problems of which the objective function is unknown and thus has to be debunked using empirical evidence. In contrast to the common practice that relies on a learning-and-optimization strategy, we consider the regression between combinatorial spaces, aiming to infer high-quality optimization solutions from samples of input-solution pairs – without the need to learn the objective function. Our main deliverable is a universal solver that is able to handle abstract undetermined stochastic combinatorial optimization problems. For learning foundations, we present learning-error analysis under the PAC-Bayesian framework using a new margin-based analysis. In empirical studies, we demonstrate our design using proof-of-concept experiments, and compare it with other methods that are potentially applicable. Overall, we obtain highly encouraging experimental results for several classic combinatorial problems on both synthetic and real-world datasets.

## 1 Introduction

Combinatorial optimization problems are not only of great theoretical interest but also central to enormous applications. Traditional research assumes that the problem settings (i.e., objective function) are completely known [1], but the reality is that the underlying system is often very complicated and only partial knowledge (from historical data) is provided. In a recommendation system, the item-user similarities could be unknown [2], which makes it impossible to compute the optimal recommendation scheme. In the study of wireless communications, the backbone network depends on stochastic node connections [3], incurring extra difficulties in designing management strategies. In the most general sense, we can formalize such scenarios by assuming that the objective function is governed by a *configuration space* associated with an *unknown* distribution, where our goal is to maximize the expected objective. For example, the configuration space may specify possible item-user similarities or candidate network structures. We call such problems as *undetermined stochastic combinatorial optimization (USCO) problems*.

Since the distribution of the configuration space is unknown, one can adopt the learning-and-optimization strategy [4], where the unknown distribution is first learned from data so that the subsequent optimization problem can be solved using existing methods. While such a strategy is natural and proved successful in several applications [5], it may require a large amount of data to learn an excessive number of parameters – for example, the weight between each pair of nodes in a network or the similarity between each pair of user and item in a recommendation system. In the worst case, we may not even have access to such kind of data (due to, for example, privacy issues

35th Conference on Neural Information Processing Systems (NeurIPS 2021).

[6]). Furthermore, a more technical concern is that the learning process is often independent of the optimization process, causing the situation that optimizing the learned objective function does not produce a good approximation to the true objective function, which is theoretically possible [7]. These concerns motivate us to think of the settings that can eschew model learning and can address USCO problems aiming right at the approximation guarantees. To this end, we consider the regression between the input space and solution space, and wish to directly compute the solution for future inputs without learning the hidden objective function.

**USCO-Solver.** In this paper, we present USCO-Solver, a general-purpose solver to USCO problems. Our framework is designed through two main techniques: randomized function approximation and approximate structured prediction. The key idea is to construct a hypothesis space composed of affine combinations of combinatorial kernels generated using random configurations, from which a large-margin machine is trained using input-solution pairs via approximate inference. The main advantage of USCO-Solver is that it can handle an abstract USCO problem as long as an oracle to solve its deterministic counterpart is given. Another significance of USCO-Solver lies in our use of combinatorial kernels, which suggests a novel and principled way for incorporating an approximation algorithm into a learning process. In doing so, we are able to handle combinatorial objects easily without worrying much about their indifferentiability or the inherited symmetries, which is often not possible for mainstream learning methods [8].

**Theoretical analysis.** For USCO-Solver, we present a learning-error analysis under the PAC-Bayesian framework, where we introduce the multiplicative margin dedicated to bounding the approximation guarantees. In particular, we prove that the approximation ratio of the predictions is essentially bounded by $O(\alpha^2)$, whenever the considered problem admits an $\alpha$-approximation in the deterministic sense. Such a result is possible due to the fact the hypothesis space of USCO-Solver can approximate the configuration space arbitrarily well. To our knowledge, this is the first result of such kind.

**Empirical studies.** We conduct experiments with USCO problems concerning three classic combinatorial objects: path, coverage, and matching. On various real-world datasets, we consistently observe that USCO-Solver not only works the way it is supposed to, but also outperforms other competitors by an evident margin in most cases. In many case studies, near-optimal solutions can be computed without consulting the true configuration distribution.

**Supplementary material.** The proofs, together with more results and discussions on the experiments, can be found in the supplementary material. In addition, we release the experimental materials, including source code, datasets, and pretrain models, as well as their instructions. The experiment materials can be found online[1].

## 2 Preliminaries

### 2.1 Undetermined stochastic combinatorial optimization problems

We are concerned with an abstract combinatorial optimization problem associated with three finite combinatorial spaces: the *input space* $\mathcal{X}$, the *output space* $\mathcal{Y}$, and a *configuration space* $\mathcal{C}$; in addition, we are given a bounded non-negative function $f(x, y, c) : \mathcal{X} \times \mathcal{Y} \times \mathcal{C} \to \mathbb{R}^+$ denoting the objective value to be maximized. Let $\Phi$ be the set of all distributions over $\mathcal{C}$. We consider *stochastic* combinatorial optimization problem in the sense that we seek to maximize $f$ in terms of a distribution $\phi_{true} \in \Phi$ rather than a fixed configuration in $\mathcal{C}$. This is desired because the system that defines the objective values is often essentially probabilistic, which may stem from random networks or functions with random parameters. Therefore, the objective function is given by

$$F(x, y, \phi_{true}) = \int_{c \in \mathcal{C}} \phi_{true}(c) \cdot f(x, y, c) \, dc. \tag{1}$$

Given $x$ and $\phi_{true}$, we are interested in the problem

$$\max_{y \in \mathcal{Y}} F(x, y, \phi_{true}). \tag{2}$$

We may wish to compute either the optimal solution $H(x, \phi_{true}) \coloneqq \arg\max_{y \in \mathcal{Y}} F(x, y, \phi_{true})$ or its $\alpha$-approximation $H_\alpha(x, \phi_{true})$ for some $\alpha \in (0, 1)$. Such problems are fundamental and also

---

[1] https://github.com/cdslabamotong/USCO-Solver

ubiquitous, ranging from the stochastic version of classic combinatorial problems [9] to applications subject to environmental uncertainties, such as commercial recommendation [10], viral marketing [11], and behavioral decision making in autonomous systems [12]. Taking the stochastic longest path problem [13] as an example, the length of each edge could follow a certain distribution, and therefore, each configuration $c \in \mathcal{C}$ corresponds to a weighted graph; in such a case, $x = (u, v)$ denotes a pair of source and destination, $y$ denotes a path from $u$ to $v$, and $f(x, y, c)$ amounts to the length of $y$ in $c$.

While the above formulation is natural, we are always limited by our imperfect understanding of the underlying system, which could be formalized by assuming that the distribution $\phi_{true}$ over the configuration is not known to us. Since the true distribution $\phi_{true}$ is unknown, the objective function $F(x, y, \phi_{true})$ cannot be directly optimized. Essentially, a learning process is required. In such a sense, we call these problems *undetermined* stochastic combinatorial optimization (USCO) problems.

**Oracle.** In general, USCO problems are made difficult by a) the learning challenges in dealing with the unknown distribution $\phi_{true}$ and b) the approximation hardness in solving Equation 2. We focus on the learning challenges and thus assume the approximation hardness (if any) can be resolved by oracles. Notice that the objective function (Equation 1) lies in the positive affine closure of its discretizations, which is

$$\cup_{k=1}^{\infty} \Big\{ \sum_{i=1}^{k} w_i \cdot f(x, y, c_i) : w_i \in \mathbb{R}^+, c_i \in \mathcal{C} \Big\}. \tag{3}$$

For some fixed $\alpha \in (0, 1]$, we assume the access to a polynomial oracle for computing an $\alpha$-approximation to maximizing any function in the above class, which implies that $H_\alpha(x, \phi)$ is obtainable for each $x \in \mathcal{X}$ and $\phi$. We can think of $\alpha$ as the best ratio that one can obtain by a polynomial algorithm, with $\alpha = 1$ denoting the scenario when the optimal solution can be efficiently computed.

For a vector $\mathbf{w} = (w_1, ..., w_k) \in \mathbb{R}^k$ and a subset $C = \{c_1, ..., c_k\} \subseteq \mathcal{C}$ of configurations, we denote the induced kernel function as

$$\mathcal{K}_C(x, y) := \Big( f(x, y, c_1), ..., f(x, y, c_k) \Big)$$

and the associated affine combination as

$$\hat{F}_{\mathbf{w}, C}(x, y) := \sum_{i=1}^{k} w_i \cdot f(x, y, c_i) = \mathbf{w}^\mathsf{T} \mathcal{K}_C(x, y).$$

Finally, we denote the $\alpha$-approximation to $\max_{y \in \mathcal{Y}} \hat{F}_{\mathbf{w}, C}(x, y)$ as $h_{\mathbf{w}, C}^\alpha(x)$.

## 2.2 Learning settings

We consider samples of input-solution pairs:

$$S_m := \Big\{ (x_i, y_i^\alpha) \Big\}_{i=1}^{m} \subseteq 2^{\mathcal{X} \times \mathcal{Y}}$$

where $y_i^\alpha := H_\alpha(x_i, \phi_{true})$ is an $\alpha$-approximation associated with the input $x_i$. Such samples intuitively offer the historical experience in which we successfully obtained good solutions to some inputs. From a learning perspective, we have formulated a regression task between two combinatorial spaces $\mathcal{X}$ and $\mathcal{Y}$. Some formal treatments are given as follows.

Let us assume that the true distribution $\phi_{true}$ is unknown but fixed, and there is a distribution $\mathcal{D}_{\mathcal{X}}$ over $\mathcal{X}$. Our goal is to design a learning framework $A_S : \mathcal{X} \to \mathcal{Y}$ that can leverage a set $S$ of samples to compute a prediction $A_S(x) \in \mathcal{Y}$ for each future $x \in \mathcal{X}$. For a prediction $\hat{y} \in \mathcal{Y}$ associated with an input $x$, let $l(x, \hat{y}) \in [0, 1]$ be a general loss function. Since we aim to examine if $\hat{y}$ is a good approximation to Equation 2, the loss $l$ is determined jointly by $\hat{y}$ and $x$. Consequently, we measure the performance of $A$ by

$$\mathcal{L}(A_S, \mathcal{D}_{\mathcal{X}}, l) := \mathbb{E}_{x \sim \mathcal{D}_{\mathcal{X}}} \big[ l(x, A_S(x)) \big]$$

## 3 A learning framework

In this section, we first present the learning framework and then analyze its generalization bounds. Finally, we discuss training algorithms.

## 3.1 USCO-Solver

Following the standard idea of structured prediction [14, 15, 16], we wish for a score function $\hat{F}(x, y)$ such that, for each $x \in \mathcal{X}$, the $y \in \mathcal{Y}$ that can maximize $\hat{F}(x, y)$ is a good solution to maximizing $F(x, y, \phi_{true})$. Suppose that we are provided with a set $S_m$ of $m \in \mathbb{Z}$ samples. USCO-Solver consists of the following steps:

- **Step 1.** Select a distribution $\phi_{em} \in \Phi$ over $\mathcal{C}$ and decide a hyperparameter $K \in \mathbb{Z}$.
- **Step 2.** Sample $K$ configurations $C_K = \{c_1, ..., c_K\} \subseteq \mathcal{C}$ independently following $\phi_{em}$.
- **Step 3.** Compute a *seed vector* $\widetilde{\mathbf{w}} = (\widetilde{w}_1, ..., \widetilde{w}_K) \in \mathbb{R}^K$ using the training data.
- **Step 4.** Sample a vector $\overline{\mathbf{w}} \in \mathbb{R}^K$ from $Q(\mathbf{w}|\beta \cdot \widetilde{\mathbf{w}}, \mathcal{I})$, which is an isotropic Gaussian with identity convariance and a mean of $\beta \cdot \widetilde{\mathbf{w}}$, with $\beta$ being

$$\beta := \frac{4}{\min_p |\widetilde{w}_p| \cdot \alpha^2} \sqrt{2 \ln \frac{2mK}{\|\widetilde{\mathbf{w}}\|^2}}. \tag{4}$$

- **Score function.** With $C_K$ and $\overline{\mathbf{w}} = (\overline{w}_1, ..., \overline{w}_K)$, we adopt the score function

$$\hat{F}_{\overline{\mathbf{w}}, C_K}(x, y) := \sum_{i=1}^{K} \overline{w}_i \cdot f(x, y, c_i).$$

- **Inference.** For each input $x \in \mathcal{X}$, the prediction is given by $h^\alpha_{\overline{\mathbf{w}}, C_K}(x)$, which can be computed by the oracle.

Given that the framework is randomized, the error of interest is

$$\mathcal{L}(\text{USCO-Solver}_{S_m}, \mathcal{D}_\mathcal{X}, l) := \mathbb{E}_{\overline{\mathbf{w}} \sim Q, C_K \sim \phi_{em}, x \sim \mathcal{D}_X} \left[ l\left(x, h^\alpha_{\overline{\mathbf{w}}, C_K}(x)\right) \right]. \tag{5}$$

So far, we have not seen how to determine the distribution $\phi_{em}$, the parameter $K$, and the seed vector $\widetilde{\mathbf{w}}$. We will first analyze how they may affect the generalization bound (Sec. 3.2), and then discuss how to make selections towards minimizing the generalization bound (Sec. 3.3).

## 3.2 Generalization bound

We first study the general loss functions and then consider a particular loss function for measuring the approximation effect in terms of Equation 2.

### 3.2.1 General loss

The generalization error in general can be decomposed into training error and approximation error [17]; the PAC-Bayesian framework gives a particular bound of such kind [18, 19]. For USCO-Solver, since the decision boundary is probabilistic (as $\overline{\mathbf{w}}$ is sampled from the seed vector $\widetilde{\mathbf{w}}$), the training error has to be bounded by considering the amount by which $h^\alpha_{\overline{\mathbf{w}}, C_K}(x_i)$ can deviate from $h^\alpha_{\widetilde{\mathbf{w}}, C_K}(x_i)$ for each training input $x_i$. To this end, we measure the similarity of two outputs $y_1, y_2 \in \mathcal{Y}$ through the concept of margin, which is given by

$$\text{m}(\mathbf{w}, C_K, x, y_1, y_2) := \alpha \cdot \hat{F}_{\mathbf{w}, C_K}(x, y_1) - \hat{F}_{\mathbf{w}, C_K}(x, y_2). \tag{6}$$

For an input $x$, the potentially good solutions are those in

$$\text{I}(x, \mathbf{w}, C_K) := \left\{ y \in \mathcal{Y} : \text{m}(\mathbf{w}, C_K, x, h^\alpha_{\mathbf{w}, C_K}(x), y) \leq \frac{\alpha}{2} \cdot \hat{F}_{\mathbf{w}, C_K}(x, h^\alpha_{\mathbf{w}, C_K}(x)) \right\}. \tag{7}$$

Intuitively, these are the solutions that are similar to the suboptimal one $h^\alpha_{\mathbf{w}, C_K}(x)$ in terms of the score function associated with $\mathbf{w}$ and $C_K$. For each $\mathbf{w}$, the training error is then taken by the worst-case loss over the possible $y$ within the margin, which is

$$\mathcal{L}(\mathbf{w}, C_K, S_m) := \frac{1}{m} \sum_{i=1}^{m} \max_{y \in \text{I}(x_i, \mathbf{w}, C_K)} l(x_i, y).$$

Notice that the loss $\mathcal{L}(\widetilde{\mathbf{w}}, C_K, S_m)$ associated with the seed vector $\widetilde{\mathbf{w}}$ can be used to bound the training error with respect to $\overline{\mathbf{w}}$, provided that the predictions are within the margin (with high probability). With such intuitions, we have the first learning bound.

**Theorem 1.** *For each $\widetilde{\mathbf{w}}$, $C_K$, and $\delta > 0$, with probability at least $1 - \delta$, we have*

$$\mathcal{L}(USCO\text{-}Solver_{S_m}, \mathcal{D}_{\mathcal{X}}, l) \leq \mathcal{L}(\widetilde{\mathbf{w}}, C_K, S_m) + \frac{\|\widetilde{\mathbf{w}}\|^2}{m} + \sqrt{\frac{\ln \frac{2Km}{\|\widetilde{\mathbf{w}}\|^2}\left(\frac{4\|\widetilde{\mathbf{w}}\|}{\min_p |\widetilde{w}_p| \cdot \alpha^2}\right)^2 + \ln \frac{m}{\delta}}{2(m-1)}}$$

The proof is inspired by the PAC-Bayesian framework [20, 21] applying to approximate structured prediction [22, 19], where our contribution lies in the analysis of the multiplicative margin.

### 3.2.2 Approximation loss

In line with Equation 2, one natural loss function is the approximation ratio of the predictions, which is

$$l_{\text{approx}}(x, \hat{y}) := 1 - \min\left(\frac{F(x, \hat{y}, \phi_{true})}{F(x, H_\alpha(x, \phi_{true}), \phi_{true})}, 1\right) \in [0, 1]. \tag{8}$$

Essentially, we seek to generalize the approximation guarantee from the training samples to future inputs, and the guarantees on the training samples are limited by the oracle. With such, we hope to compare the generalization error under $l_{\text{approx}}$ with $\alpha$. According to Theorem 1, with an unlimited supply of training data (i.e., $m \to \infty$), the generalization error is bounded by $\mathcal{L}(\widetilde{\mathbf{w}}, C_K, S_m)$, and therefore, the learning error of the ERM solution is concentrated on

$$\inf_{\widetilde{\mathbf{w}}} \mathbb{E}_{x \sim \mathcal{D}_{\mathcal{X}}} \left[ \max_{y \in \mathrm{I}(x_i, \widetilde{\mathbf{w}}, C_K)} l_{\text{approx}}(x_i, y) \right]. \tag{9}$$

Relating the above quantity with $\alpha$ is not possible for the general case because the score function $\hat{F}_{\widetilde{\mathbf{w}}, C_K}(x, y)$ defining the margin is independent of the objective function $F(x, y, \phi_{true})$ inducing the error. However, more concrete results can be obtained for the approximation loss, by leveraging a subtle relationship between our construction of the kernel function and the stochastic combinatorial optimization problem. The next result characterizes the relationship between Equation 9 and $\alpha$.

**Theorem 2.** *Suppose that $f(x, y, c) \in [A, B]$ and $C := \sup_c \frac{\phi_{true}(c)}{\phi_{em}(c)}$. For each $\epsilon > 0, \delta_1 > 0, \delta_2 > 0$ and $\phi_{em}$, when $K$ is no less than*

$$\frac{2 \cdot C^2 \cdot B^2}{\epsilon^2 \cdot \delta_2^2 \cdot A^2} \cdot \max\left(\frac{1}{2}, \ln |Y| + \ln \frac{1}{\delta_1}\right) \tag{10}$$

*with probability at least $1 - \delta_1$ over the selection of $C_K$, there exists a $\widetilde{\mathbf{w}}$ such that*

$$\Pr_{x \sim \mathcal{D}_{\mathcal{X}}} \left[ \max_{y \in \mathrm{I}(x_i, \widetilde{\mathbf{w}}, C_K)} l_{\text{approx}}(x_i, y) \leq \frac{(1 + \epsilon) - (1 - \epsilon)(\alpha^2/2)}{(1 + \epsilon)} \right] \geq 1 - \delta_2.$$

The above result shows that the approximation ratio in generalization is essentially bounded by $\alpha^2/2$. It is intuitive that the bound on $K$ depends on the deviation of $\phi_{em}$ from $\phi_{true}$ (i.e., $C$), the range of the objective function (i.e.. $B/A$), and the size of the output space.

### 3.3 Training algorithms

Theorems 1 and 2 have explained how $m$, $K$ and $\phi_{em}$ may affect the generalization performance in theory. Practically, $K$ can be conveniently taken as a hyperparameter, and $\phi_{em}$ is often the uniform distribution over $\mathcal{C}$ (unless prior knowledge about $\phi_{true}$ is given). Now, for the four steps in Sec. 3.1, the only problem left is to compute the seed vector $\widetilde{\mathbf{w}}$.

**Relaxed margin.** While Theorem 2 indicates that there exists a desired weight, we are not able to search for such a weight directly because the loss function $l_{\text{approx}}$ is not accessible under our setting. In practice, one can adopt the zero-one loss or other loss function preferred by the considered problem. For example, when the output space is a family of sets, the loss function can be induced by the set similarity [23]. For a general loss function $l(x_i, y)$, Theorem 1 suggests computing the weight $\widetilde{\mathbf{w}}$ by minimizing the upper bound $\mathcal{L}(\widetilde{\mathbf{w}}, C_K, S_m) + \frac{\|\widetilde{\mathbf{w}}\|^2}{m}$:

$$\min_{\mathbf{w}} \max_{y} l(x_i, y) \cdot \mathbb{1}(y \in \mathrm{I}(x_i, \mathbf{w}, C_K)) + ||\mathbf{w}||^2,$$

where $\mathbb{1}$ is the indicator function. Notice that the above problem is not only non-convex but also complicated by the fact that $\mathbf{w}$ and $h^\alpha_{\mathbf{w}, C_K}(x)$ are convoluted with each other. For ease of optimization, a relaxed margin will be useful, as seen shortly. Notice that we have $\hat{F}_{\mathbf{w}, C_K}(x_i, h^\alpha_{\mathbf{w}, C_K}(x_i)) \geq \alpha \cdot \hat{F}_{\mathbf{w}, C_K}(x_i, y^\alpha_i)$, and therefore, for each $y \in \mathcal{Y}$, $y \in \mathrm{I}(x_i, \mathbf{w}, C_K)$ implies that

$$y \in \bar{\mathrm{I}}(x, \mathbf{w}, C_K) := \left\{ y \in \mathcal{Y} : \frac{\alpha^2}{2} \cdot \hat{F}_{\mathbf{w}, C_K}(x_i, y^\alpha_i) - \hat{F}_{\mathbf{w}, C_K}(x_i, y) \leq 0 \right\},$$

where the new margin $\bar{\mathrm{I}}$ is measured with respect to $y^\alpha_i$ instead of $h^\alpha_{\mathbf{w}, C_K}(x_i)$. Immediately, we have $\mathrm{I}(x_i, \mathbf{w}, C_K) \subseteq \bar{\mathrm{I}}(x_i, \mathbf{w}, C_K)$, which indicates that the error is further upper bounded by

$$\max_y l(x_i, y) \cdot \mathbb{1}(y \in \bar{\mathrm{I}}(x_i, \mathbf{w}, C_K)) + ||\mathbf{w}||^2, \tag{11}$$

which is less challenging to minimize.

**Large-margin training.** Seeking a large-margin solution and scaling the margin proportional to the loss function, Equation 11 amounts to solving the following problem:

$$\min \quad \frac{1}{2} \|\mathbf{w}\|^2 + \frac{C}{2m} \sum_{i=1}^m \xi_i$$

$$\text{s.t.} \quad \frac{\alpha^2}{2} \mathbf{w}^\mathsf{T} \mathcal{K}_{C_K}(x_i, y^\alpha_i) - \mathbf{w}^\mathsf{T} \mathcal{K}_{C_K}(x_i, y) \geq \eta \cdot l(x_i, y) - \xi_i, \ \forall i \in [m], \ \forall y \in \mathcal{Y}, y \neq y^\alpha_i$$

$$\mathbf{w} \geq 0$$

where $C$ and $\eta$ are hyperparameters. Notably, this is the vanilla structured SVM [16, 24]. Viewing $\mathcal{K}_{C_K}$ as a score function, the constrains enforces that the solution $y^\alpha_i$ provided by the training data should have a high score. The above formulation appears to be a standard quadratic programming, but it in fact contains $|\mathcal{Y}|$ number of constrains, which can be prohibitively large (e.g., exponential in the coding length of the configuration). Since the constraints are equivalent to

$$\max_{y \in \mathcal{Y}} \mathbf{w}^\mathsf{T} \mathcal{K}_{C_K}(x_i, y) + \eta \cdot l(x_i, y) \leq \frac{\alpha^2}{2} \mathbf{w}^\mathsf{T} \mathcal{K}_{C_K}(x_i, y^\alpha_i) + \xi_i, \ \forall i \in [m],$$

the number of constraints could be reduced as linear in sample size if we can effectively solve the loss-augmented inference problem:

$$\max_{y \in \mathcal{Y}} \mathbf{w}^\mathsf{T} \mathcal{K}_{C_K}(x_i, y) + \eta \cdot l(x_i, y)$$

For the zero-one loss, this is exactly to solve $\max_{y \in \mathcal{Y}} \mathbf{w}^\mathsf{T} \mathcal{K}_{C_K}(x_i, y)$, which, to our delight, can be solved using the oracle in Sec. 2.1. Once such loss-augmented inference problem can be solved, the entire programming can be readily solved by existing methods, such as the cutting-plane algorithm [25]. This completes the learning framework.

**Remark 1.** While our discussion so far is focused on maximization problems, the theoretical analysis can be adapted to minimization problems easily. In the training part for minimization problems, the major adaption is to reverse $\mathbf{w}^\mathsf{T} \mathcal{K}_{C_K}(x_i, y^\alpha_i)$ and $\mathbf{w}^\mathsf{T} \mathcal{K}_{C_K}(x_i, y)$, which also turns the loss-augmented inference into a minimization problem.

**Remark 2.** For the stochastic shortest path problem, $\phi_{em}$ denotes a distribution over a collection $\mathcal{C}$ of weighted graphs, and assuming the weights are nonnegative, we may use Dijkstra algorithm as the oracle, as $\max_{y \in \mathcal{Y}} \hat{F}_{\mathbf{w}, C}(x, y)$ amounts to computing the shortest path in the combining graph of the sampled weighted graphs.

## 4 Empirical studies

This section presents experiments across a variety of combinatorial problems, including stochastic shortest path (SSP), stochastic set cover (SSC), and stochastic bipartite matching (SBM). In addition to supporting the theoretical analysis, our empirical studies are of interest also because none of the existing methods is known to be effective for solving any USCO problem.

Table 1: **SSP results.** Each cell shows the mean of performance ratio with the standard deviation.

| | | UCSO-Solver | | | | | Other Methods | | |
|---|---|---|---|---|---|---|---|---|---|
| | $K$ | 16 | 160 | 1600 | 3200 | 6400 | NB | DSPN | Base |
| **Col** | $\phi_{exp}$ | 1.942 (0.11) | 1.952 (0.04) | 1.565 (0.02) | 1.129 (0.01) | 1.148 (0.01) | 2.064 (0.16) | 2.009 (0.12) | 1.956 (0.10) |
| | $\phi_{true}$ | 1.300 (0.02) | 1.015 (0.01) | 1.016 (0.01) | 1.010 (0.01) | 1.022 (0.01) | | | |
| | $K$ | 80 | 160 | 640 | 3200 | 6400 | NB | DSPN | Base |
| **NY** | $\phi_{exp}$ | 1.613 (0.01) | 1.571 (0.02) | 1.353 (0.01) | 1.303 (0.02) | 1.192 (0.01) | 3.155 (0.03) | 2.469 (0.07) | 2.853 (0.28) |
| | $\phi_{true}$ | 1.205 (0.01) | 1.177 (0.01) | 1.017 (0.01) | 1.031 (0.01) | 1.048 (0.01) | | | |
| | $K$ | 160 | 1600 | 3200 | 6400 | 9600 | NB | DSPN | Base |
| **Kro** | $\phi_{exp}$ | 7.599 (0.15) | 5.829 (0.22) | 5.858 (0.18) | 4.613 (0.18) | 4.323 (0.19) | 10.08 (0.57) | 7.451 (1.05) | 10.92 (2.45) |
| | $\phi_{true}$ | 1.361 (0.09) | 1.026 (0.01) | 1.022 (0.01) | 1.042 (0.01) | 1.051 (0.01) | | | |

**Evaluation metric.** For each training pair $(x_i, y_i^\alpha)$ and a prediction $\hat{y}$, the performance ratio is defined as $\frac{F(x_i, y_i^\alpha, \phi_{true})}{F(x_i, \hat{y}, \phi_{true})}$ for maximization problems and $\frac{F(x_i, \hat{y}, \phi_{true})}{F(x_i, y_i^\alpha, \phi_{true})}$ for minimization problems. Therefore, lower is better in both cases. The performance is measured over five random testings. We here present the main experimental settings and discuss key observations, deferring the complete analysis to Appendix B.

## 4.1 Stochastic shortest path (SSP)

**Problem definition and oracle.** In the stochastic version of the shortest path problem, given a source $u$ and a destination $v$ in a graph $G = (V, E)$, we wish to find the path (from $u$ to $v$) that is the shortest in terms of a distribution over the possible edge weights. In this case, the configuration space $\mathcal{C}$ denotes a family of weighted graphs associated with a distribution $\phi_{true}$, the input $x = (u, v)$ is a pair of nodes, the output $y$ is a path from $u$ and $v$, and $f(x, y, c)$ denotes the length of $y$ in graph $c$. We construct instances where the edge weights are non-negative, and thus, each function in Equation 3 can be minimized optimally using the Dijkstra algorithm [26], which is the oracle we use to generate samples and solve the inference problem in USCO-Solver.

**Instance construction.** To setup a concrete instance, we first fix a graph structure, where two real-world graphs (Col and NY) and one synthetic graph (Kro) are adopted. Col and NY are USA road networks complied from the benchmarks of DIMACS Implementation Challenge [27], and Kro is a Kronecker graph [28]. Following the common practice [29], we assign each edge a Weibull distribution with parameters randomly selected from $\{1, ..., 10\}$, which – together with the graph structure – induces the configuration space $\mathcal{C}$ and $\phi_{true}$. For each instance, we generate the pool of all possible input-solution pairs $(x, y)$, where, for an input $x = (u, v)$, $y$ is the optimal solution to $\min_{y \in \mathcal{Y}} F(x, y, \phi_{true})$. For each considered method, we randomly select 160 training samples and 6400 testing samples from the pool.

**Implementing USCO-Solver.** We setup two realizations ($\phi_{exp}$ and $\phi_{true}$) of $\phi_{em}$ that will be used in USCO-Solver. $\phi_{exp}$ assigns each edge in $E$ an exponential distribution normalized in $[1, 1e5]$, and $\phi_{true}$ is exactly the true underlying distribution that defines the instance. Notice that $\phi_{true}$ is not obtainable under our learning setting, and it is used only to verify our designs. For each distribution, we sample a pool of 10000 configurations (i.e., weighted graphs). The number of configurations (i.e., $K$) is enumerated from small to large, with different ranges for different datasets. Given the size $K$, the configurations are randomly selected from the pool in each run of USCO-Solver. We use the zero-one loss, and the training algorithm is implemented based on Pystruct [30].

**Implementing other competitors.** For a baseline, we implement the Base method which, given an input $x = (u, v)$, outputs the shortest path from $u$ to $v$ in $G$ where the edge weights are randomly sampled from $[0, 1]$. Conceptually, our problem can be perceived as a supervised learning problem from $V \times V$ to the space of the paths, and thus, off-the-shelf learning methods appear to be applicable, which however turns out to be non-trivial. We managed to implement two methods based respectively on Naive Bayes (NB) [31] and Deep Set Prediction Networks (DSPN) [32], where the idea is to

Table 2: **SSC results.** The table shows the results on two datasets Cora and Yahoo.

| | | UCSO-Solver | | | | | Other Methods | | |
|---|---|---|---|---|---|---|---|---|---|
| | $K$ | 8 | 16 | 160 | 320 | 640 | GNN | DSPN | Rand |
| **Cora** | $\phi_{uni}$ | 1.480 (0.12) | 1.452 (0.12) | 1.230 (0.06) | 1.147 (0.06) | 1.083 (0.01) | 1.036 (0.02) | 1.782 (0.85) | 16.57 (0.42) |
| | $\phi_{true}$ | 1.029 (0.01) | 1.033 (0.01) | 1.003 (<0.01) | 1.000 (<0.01) | 1.000 (<0.01) | | | |
| | $K$ | 8 | 16 | 160 | 320 | 640 | GNN | DSPN | Rand |
| **Yahoo** | $\phi_{uni}$ | 1.294 (0.03) | 1.356 (0.03) | 1.151 (0.01 | 1.146 (0.07) | 1.093 (0.03) | 1.076 (0.10) | 1.387 (0.142) | 6.58 (0.17) |
| | $\phi_{true}$ | 1.036 (0.02) | 1.013 (0.01) | 1.003 (<0.01) | 1.009 (<0.01) | 0.999 (<0.01) | | | |

leverage them to learn the node importance in the shortest path and then build the prediction based on the node importance.

**Observations.** The main results are presented in Table 1. We highlight two important observations: a) the performance of USCO-Solver smoothly increases when more configurations are provided, and b) when fed with configurations from the true distribution $\phi_{true}$, USCO-Solver can easily output near-optimal solutions on all the datasets. These observations confirm that USCO-Solver indeed functions the way it is supposed to. In addition, the efficacy of USCO-Solver is significant and robust in terms of the performance ratio. On datasets like Col and NY, it can produce near-optimal solutions using $\phi_{exp}$. Meanwhile, other methods are not comparable to USCO-Solver, though DSPN offers non-trivial improvement compared with Base on NY and Kro. It is also worth noting that different instances exhibit different levels of hardness with respect to the number of configurations needed by USCO-Solver. 3200 configurations are sufficient for USCO-Solver to achieve a low ratio on Col, while the ratio is still larger than $4.0$ on Kro even though 9600 configurations have been used. See Appendix B.2 for a complete discussion.

### 4.2 Stochastic set cover (SSC)

In network science, coverage problems often require to compute a set of terminals that can maximally cover the target area [33]; in applications like recommendation system or document summarization [34], one would like to select a certain number of items that can achieve the maximum diversity by covering topics as many as possible. These problems can be universally formulated as a maximal coverage problem [35], where each instance is given by a family $U$ of subsets of a ground set, where our goal is to select a $k$-subset of $U$ of which the union is maximized. In its stochastic version, we assume that an item will appear in a subset with a certain probability [36]. The oracle we use is the greedy algorithm, which is a $(1 - 1/e)$-approximation [37]. We construct two instances based on two real-world bipartite graphs Cora [38] and Yahoo [39]. Similar to the setup for SSP, we first generate the ground truth distribution $\phi_{true}$ and then generate samples. We implement two learning methods based on Graph neural network (GNN) and DSPN, together with a baseline method Rand that randomly selects the nodes in $\hat{y}$. For USCO-Solver, we consider a distribution $\phi_{uni}$ that generates a configuration by generating subsets uniformly at random. See Appendix B.3 for details.

The results are presented in Table 2. First, the observations here are similar to those in the SSP problem – USCO-Solver works in a manner as we expect, and it can produce high-quality solutions when a sufficient number of configurations are provided. We can see that GNN and DSPN are also effective compared with Rand, which suggests that, compared to SSP, SSC is much easier for being handled by existing learning methods.

### 4.3 Stochastic bipartite matching (SBM)

Our last problem is bipartite matching, which is regularly seen in applications such as public housing allocation [40], semantic web service [41] and image feature analysis [42]. We consider the minimum weight bipartite matching problem. Given a weighted bipartite graph $G = (L, R, E)$, the input $x = (L^*, R^*)$ consists of two subsets $L^* \subseteq L$ and $R^* \subseteq R$ with $|L^*| = |R^*|$, and the output $y$ is a perfect matching between $L^*$ and $R^*$ such that the total cost is minimized. In other words, $y$ is a bijection between $L^*$ and $R^*$. In its stochastic version, the weight $w_e$ for each edge $e \in E$ follows a certain distribution, and we aim to compute the matching that can minimize the expected

Table 3: **SBM results with $\phi_{uni}$ and $\phi_{true}$.**

| $K$ | 16 | 160 | 1600 | 3200 | 6400 | 12800 | 19200 | Rand |
|---|---|---|---|---|---|---|---|---|
| $\phi_{uni}$ | 3.627 (0.01) | 3.582 (0.01) | 3.407 (0.01) | 3.261 (0.01) | 3.107 (0.01) | 2.874 (0.01) | 2.696 (0.01) | 3.670 (0.01) |
| $\phi_{true}$ | 1.029 (0.01) | 1.033 (0.01) | 1.003 (<0.01) | 1.000 (<0.01) | 1.000 (<0.01) | | | |

Table 4: **SBM results with $\phi_{10}$, $\phi_5$, $\phi_1$ and $\phi_{0.3}$.**

| | 16 | 160 | 320 | 640 | | 16 | 160 | 320 | 640 |
|---|---|---|---|---|---|---|---|---|---|
| $\phi_{10}$ | 4.889 (0.09) | 3.602 (0.06) | 1.831 (0.04) | 1.271 (0.01) | $\phi_5$ | 4.321 (0.07) | 1.403 (0.01) | 1.120 (0.01) | 1.040 (<0.01) |
| $\phi_1$ | 1.038 (<0.01) | 1.008 (<0.01) | 1.003 (<0.01) | 1.002(<0.01) | $\phi_{0.3}$ | 1.008 (<0.01) | 1.003 (<0.01) | 1.002 (<0.01) | 1.002(<0.01) |

cost $F(x, y, \phi_{true})$. Notice that the minimum weight bipartite matching problem can be solved in polynomial time by linear programming [43], which is a natural oracle to use in USCO-Solver. We adopt a synthetic bipartite graph with 128 nodes. The weight $w_e$ for each edge $e \in E$ follows a Gaussian distribution $\mathcal{N}(\mu_e, \sigma_e)$ with $\mu_e$ sampled uniformly from $[1, 10]$ and $\sigma_e = 0.3 \cdot \mu_e$, which induces the true distribution $\phi_{true}$. See Appendix B.4 for details.

**Results under $\phi_{uni}$.** For USCO-Solver, we consider $\phi_{uni}$ that generates the configuration $c$ by giving each edge a weight from $[1, 10]$ uniformly at random. We also test USCO-Solver with features from the true distribution $\phi_{true}$. The baseline method Rand produces $y$ for a given input $x = (L^*, R^*)$ by randomly generating a permutation of $R^*$. The result of this part is shown in Table 3. We see that USCO-Solver can again produce a better ratio when more configurations are provided, but different from SSP and SSC, it cannot achieve a ratio that is close to 1 even when 19200 configurations have been used. Plausibly, this is because the output space of SBM is more structured compared with SSP and SSC.

**Incorporating prior knowledge into USCO-Solver.** For problems like bipartite matching where uniform configurations cannot produce a near-optimal solution, it would be interesting to investigate if USCO-Solver can easily benefit from domain expertise. To this end, given a parameter $q \in \{0.3, 1, 5, 10\}$, we consider the distribution $\phi_q$ that samples the weight for edge $e$ from $[\mu_e - q \cdot \mu_e, \mu_e + q \cdot \mu_e]$, which accounts for the case that a confidence interval of the weight $w_e$ is known. The performance ratios produced by such distributions are shown in Table 4. As we can see from the table, the result indeed coincides with the fact that the prior knowledge is strong when $p$ is small. More importantly, with the help of such prior knowledge, the ratio can be reduced to nearly 1 using no more than 160 configurations under $\phi_1$, while under the uniform distribution $\phi_{uni}$, the ratio was not much better than Rand with the same amount of configurations (as seen in Table 3).

## 5 Further discussions

**Related work.** The learning-and-optimization framework has been used widely (e.g., [44, 45, 46, 47, 48]). For example, Du *et al.* [49] and He *et al.* [50] study the problem of learning influence function which is used in influence maximization; recently, Wilder *et al.* [51] proposes a decision-focused learning framework based on continuous relaxation of the discrete problem. In contrast to these works, we consider the input-solution regression without attempting to learn the objective function. A similar setting was adopted in [52] for studying a special application in social network analysis, but our paper targets abstract problems and provides generalization bounds on the approximation ratio. There have been recent efforts to develop learning methods that can handle combinatorial spaces (e.g., [53, 54, 32]), where the key is to preserve the inherited combinatorial structures during the learning process. Our research is different because the problem we consider involves a hidden optimization problem; in other words, we target the optimization effect rather than the prediction accuracy. Our research is also related to the existing works that attempt to solve combinatorial optimization problems using reinforcement learning [55, 56, 57, 58]; however, the objective function is known in their settings, while we study the model-free setting.

**Limitations.** Our theoretical analysis is based on the assumption that $y$ is an approximation solution in each input-solution $(x, y)$, but in practice, it may be impossible to prove such guarantees for real datasets. Thus, it is left to experimentally examine USCO-Solver using the training pairs $(x, y)$ where

$y$ is produced using heuristic but not approximation algorithms. In another issue, we have shown that USCO-Solver is effective for three classic combinatorial problems, but it remains unknown if USCO-Solver is universally effective, although it is conceptually applicable to an arbitrary USCO problem. In addition, our experiments do not rule out the possibility that off-the-shelf learning methods can be effective for some USCO problems (e.g., SSC), which suggests a future research direction. Finally, we hope to extend our research by studying more practical applications involving highly structured objects, such as Steiner trees, clustering patterns, and network flows. See Appendix C for a more comprehensive discussion.

## Acknowledgments and Disclosure of Funding

We thank the reviewers for their time and insightful comments. Funding in direct support of this work: support from the University of Delaware.

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
