# OpenReview forum: "USCO-Solver: Solving Undetermined Stochastic Combinatorial Optimization Problems"
_NeurIPS.cc/2021/Conference — NeurIPS 2021 Poster_

### Official Review · Reviewer_ToTX · 2021-07-15

**Rating:** 6
**Confidence:** 3

**Summary:**

In this paper, the authors propose USCO-Solver, a unified approach to solving stochastic combinatorial optimization by learning the weights of the kernel function. The learning step is similar to SVM. The authors conduct experiments on 3 stochastic problems.

**Limitations And Societal Impact:**

yes

**Main Review:**

Strengths of this paper:
1. A unified approach for stochastic combinatorial optimization is demanding and appealing.
1. Detailed analysis of training and generalization bounds of the proposed method.
1. Involving an oracle (polynomial-time solver) in the learning to optimization pipeline.

The limitations of this paper are mainly about the experiment part:
1. The problems considered in this paper do not fully reflect the ubiquitous challenging NP-hard combinatorial problems. The non-stochastic versions of shortest path and bipartite matching both can be solved in polynomial time. I have some concerns about the performance when generalizing USCO-Solver to more challenging combinatorial problems, especially the authors claim that they propose a general approach.
1. The baseline linear programming method [43] is missing for the stochastic bipartite matching experiment.

Some points that are unclear to me:
1. The authors write that their algorithm learns a mapping from the input space to the solution space. However, it seems that only a combination of kernels is learned, and the mapping to the solution space is done by the oracle. In my opinion, only the methods like [55] which directly predict the solution by a deep neural network can be regarded as learning a mapping from the input space to the solution space.
1. Why shall we sample $\bar{\mathbf w}$ in Step 4 of the proposed approach? It seems that the analysis in Sec 3.2 is not related to $\bar{\mathbf w}$, but mainly related to $\widetilde{\mathbf w}$.
1. The results in Table 2 seem to be computed by optimal solutions. How do you compute the optimal solution for Stochastic set cover (SSC) since the oracle is an approximate algorithm?

**Time Spent Reviewing:**

6

---

> ### Author Response · Authors · 2021-08-05
> **Author Response**
>
> We thank the reviewer for their constructive comments and questions.
>
> $\textbf{On experiments.}$ It is true that it remains unknown whether or not USCO-Solver is effective for other combinatorial optimization problems (as mentioned in Section 5), and we believe that a comprehensive experimental study is a critical following step. Our approach is general in the sense that it is applicable to abstract combinatorial optimization problems, and we look forward to seeing its practical performance for more challenging problems. Linear programming is the method we use to implement the oracle in USCO-Solver for the SBM problem, and it is not a baseline method for solving the SBM problem.
>
>
> $\textbf{The essence of the model.}$ We view our framework as a mapping from the input space to the solution space in the sense that once the weights have been learned, USCO-Solver can output a solution for any input. From a high-level perspective, the weights in USCO-Solver are analogous to the weights in the deep neural network in [55], the kernel combination in USCO-Solver is analogous to the representation learning part in [55], and the inference phase of USCO-Solver is analogous to the greedy node selection in [55] (Figure 1 in [55]). We will clarify this in the updated version.
>
> $\textbf{Relationship between $\bar{W}$ and $\tilde{W}$}$. Since $\bar{W}$ is sampled from the $Q$ induced by $\tilde{W}$, the learning outcome is essentially determined by $\tilde{W}$. It is feasible that we might use a deterministic approach and take $\bar{W}$ as the mean of $Q$ (as discussed in Chapter 31 in [a]), which is a special case of our analysis. We thank the reviewer for pointing this out, and we will add explanations regarding this issue.
>
> $\textbf{[a]}$ Shalev-Shwartz, Shai, and Shai Ben-David. Understanding machine learning: From theory to algorithms. Cambridge university press, 2014.
>
> $\textbf{Table 2.}$ We wish to note that Table 2 presents the ratio between the quality of predicted solution and quality of the (1-1/e)-approximation, without involving the optimal solutions. Intuitively, Table 2 shows that the predicted solution is comparable to the (1-1/e)-approximation, which is the best possible polynomial approximation.

---

### Official Review · Reviewer_pDuR · 2021-07-16

**Rating:** 7
**Confidence:** 3

**Summary:**

This paper proposes a learning method, USCO_Solver, to solve undetermined stochastic combinatorial optimization problems. Approximation errors of the proposed method are established, and empirical results demonstrate the success of the proposed method.

**Limitations And Societal Impact:**

Yes

**Main Review:**

This paper is well written and theoretically sound. Personally, I like the error analysis and the adoption of large-margin training. I only have two questions listed below:

1. Running time is an important aspect in solving optimization problems. How is the running time of the proposed USCO Solver compared to baselines?

2. Is it possible to train the USCO Solver on one graph (problem instance), and test it on multiple different graphs? If this can be done, the method would be more useful and of practical relevance.


**Time Spent Reviewing:**

2

---

> ### Author Response · Authors · 2021-08-05
> **Author Response**
>
> We thank the reviewer for the recognition and questions.
>
> $\textbf{Running time.}$ Theoretically, following directly from Corollary 2 in [a], for fixed hyperparameters, the time complexity of training USCO-Solver is $O(m*A+m^2+mK+K^3)$ where $m$ is the number of training instances, $A$ is the running time of the oracle (in Sec 2.1), and $K$ is the number of sampled configurations. In practice, the baseline methods (i.e., Base in SSP and Rand in SSC and SBM) are very fast, and USCO-Solver has a similar running time as other learn-based methods. Overall, the training phase of all methods is reasonably fast and within two hours. We will add such discussions to the updated version.
>
> $\textbf{[a]}$ Joachims, Thorsten, Thomas Finley, and Chun-Nam John Yu. "Cutting-plane training of structural SVMs." Machine learning 77, no. 1 (2009): 27-59.
>
> $\textbf{Transformation between graphs.}$ The current version of USCO-Solver does not feature any module for transfer learning and thus cannot make predictions on a different graph. We share the same view with the reviewer that it is interesting to explore the alignment between solutions over different graphs.

---

### Official Review · Reviewer_SRyD · 2021-07-16

**Rating:** 6
**Confidence:** 3

**Summary:**

The paper presents USCO-Solver, a new solver that is able to handle abstract undetermined stochastic combinatorial optimisation problems. The main contribution is a learning framework that allows the solver to infer high quality solutions from samples of input-output pairs without having to learn the objective function. This approach seems to contrast the well known learning-and-optimisation strategy. An empirical evaluation on several classic combinatorial problem domains demonstrate the effectiveness of the proposed approach.

**Ethical Concerns:**

There are no ethical concerns.

**Ethics Review Area:**

["I don’t know"]

**Limitations And Societal Impact:**

The limitations are addressed in the paper and there is no potential negative societal impact of this work.

**Main Review:**

The paper is fairly well written and organised. The quality of the presentation is overall fairly good. However, there are a couple of issues that I think need to be addressed.

1. The solver assumes the existence of an oracle to do the actual inference, namely to compute the prediction for each input x. This may be a serious limitation because many realistic problems it's actually very hard to compute a good quality solution. Moreover, it may actually be impossible to come up with a reliable oracle unless the decision maker knows enough about the structure of the optimisation problem. Perhaps a deeper discussion on the choice of the oracle is warranted.

2. I suggest accompanying the presentation of the learning framework in section 3 with a more concrete running example (maybe use the stochastic shortest path example from the experimental section). This would be really helpful to get a better understanding of the proposed approach.


**Time Spent Reviewing:**

3 hours

---

> ### Author Response · Authors · 2021-08-05
> **Author Response**
>
> We thank the reviewer for their suggestion and constructive comments.
>
> $\textbf{On the choice of the oracle.}$ Based on our setting, it is true that one needs to have knowledge about the structure of the optimization problem to obtain the oracle. The entire problem is made difficult by the hardness in a) computing the optimal decision (i.e., how to obtain an effective oracle) and b) overcoming the unknown configuration distribution (i.e., learning foundations). Our paper focuses primarily on the learning foundations and thus presumes the existence of a desired oracle. For applications like stochastic vehicle routing and social influence study that are formulated based on classic combinatorial optimization problems, such an oracle is relatively easy to obtain by using existing approximation techniques, but for other real-life decision-making problems, it has to be negotiated on a case-by-case basis depending on the specific choice of $f(x,y,c)$, which we believe is an interesting future work and also a key step for making USCO-Solver more practically useful.
>
> $\textbf{Concrete running example.}$ Thank you for the suggestion. We will add a concrete running example using SSP at the end of Section 3 to illustrate the proposed approach.

---

### Official Review · Reviewer_fagS · 2021-07-16

**Rating:** 5
**Confidence:** 3

**Summary:**

This paper proposes a framework for learning to predict approximate solutions to stochastic combinatorial optimization problems. The class of problems is referred to as "undetermined" stochastic combinatorial optimization. "Undetermined" here refers to the following assumption on the available training data: each data point is a pair (x, y), where x is the input to the stochastic optimization problem (e.g., source-target pairs in an s-t shortest path problem on a fixed graph), and y is an approximate solution; notice how information about the probability distribution over which the true stochastic objective function is defined (e.g., edge weights) is not available in the training set (neither via a set of sample scenarios nor a generator), hence "undetermined".

USCO-Solver does the following in the training phase: it selects an arbitrary distribution over the unknown stochastic parameters of the problems (e.g., edge weights in a graph), samples K configurations, then searches for a weight vector for the K samples such that the true solution (y) is also near-optimal in another optimization problem whose objective function is a linear combination of the K configurations' objectives. This training problem is framed as a structured prediction task which is amenable to structured SVM-style optimization algorithms. Key to the training is the use of an oracle for the deterministic counterpart of the stochastic problem.

Experiments on randomly generated datasets for stochastic shortest path, set cover, and bipartite matching show that USCO-Solver performs significantly better than "direct" ML methods that are trained to simply fit the training set without regard to the combinatorics or stochastic nature of the true optimization problem.


**Main Review:**

Assessment: I find it truly remarkable that any bounds can be derived for the proposed method, though I am unable to verify the proofs fully. The USCO setting is very challenging and USCO-Solver is simple, principled, and effective in practice. However, the USCO setting is also not realistic, which limits the impact and applicability of USCO-Solver.

Originality: There is a closely related line of work by Parmentier [Parmentier, A. (2021). Learning to approximate industrial problems by operations research classic problems. Operations Research.] which is not discussed. Although this paper appeared only recently in OR, I believe earlier versions have been online for 2+ years (https://hal.archives-ouvertes.fr/hal-02396091/). Other relevant work which could potentially be applied to USCO problems and compared to include:
- [Pogančić, Marin Vlastelica, et al. "Differentiation of blackbox combinatorial solvers." International Conference on Learning Representations. 2019.];
- [Bengio, Yoshua, et al. "A learning-based algorithm to quickly compute good primal solutions for Stochastic Integer Programs." International Conference on Integration of Constraint Programming, Artificial Intelligence, and Operations Research. Springer, Cham, 2020.]
- [Nair, Vinod, et al. "Learning Fast Optimizers for Contextual Stochastic Integer Programs." UAI. 2018.]

Clarity: The core technical parts of the paper (3.2, 3.3) are a bit difficult to follow as there many parameters that affect the bounds. Other sections are clearer.

Quality: The technical quality seems good given how difficult the problem setting is. The experimental setup is satisfactory.

Significance: As mentioned earlier, I think the main limiting factor is that the USCO setting is not realistic in two ways:
- The training data must include approximate solutions;
- One cannot sample from the true configuration distribution.

These two are somewhat contradictory, in the sense that I cannot imagine computing the approximate solutions if I cannot observe a sample from the true configuration distribution.
Without the approximate solutions, the USCO framework falls apart as there are no labels to supervise the training; if one can sample, then one can also sample at test time and use Sample Average Approximation [Kleywegt, Anton J., Alexander Shapiro, and Tito Homem-de-Mello. "The sample average approximation method for stochastic discrete optimization." SIAM Journal on Optimization 12.2 (2002): 479-502.], the standard method for stochastic optimization with expectations in the objective.

Nonetheless, I look forward to the authors' rebuttal. I think this is a crucial point.

Other comments:
- Training complexity: can you provide running time complexity for the training procedure as a function of the various hyperparameters?

- line 188: why do you only minimize the sum of the first two terms in the RHS of the inequality in Theorem 1? What happens with the third term, which is non-negative and also depends on $\widetilde{\boldsymbol{w}}$?

- line 220: "The performance is measured over ﬁve random testings."; I am not sure what you mean here, random testings with respect to which distribution?


**Time Spent Reviewing:**

4

---

> ### Author Response · Authors · 2021-08-05
> **Author Response**
>
> We appreciate the reviewer for their time and insightful comments.
>
> $\textbf{Regarding the USCO setting.}$ It is true that having the approximation solution in the dataset is somewhat contradictory against the assumption that one cannot know the true configuration distribution; we thank the reviewer for pointing this out. We wish to note that the assumption that the training data includes approximate solutions (to a certain extend) is the minimum requirement for having any performance bound, and such an assumption is introduced primarily for theoretically understanding the model performance.
>
> The overall idea behind the USCO-Solver is to directly infer good solutions for future inputs using solutions that are known to be good, and the approximation ratio is adopted to measure such goodness. In practice, USCO-Solver only expects the training set to include reasonably good solutions observed/collected by the agent. One possibility is that the underlying system is accessible during data collection but not available when the data is used later for building decision-making models. In the case that the exact approximation ratio is unknown, we could set alpha as 1.0 in real implementations, implying that the training set includes the best solution one can obtain. Our paper focuses more on the foundations of such a new learning framework, and we hope to explore more real-life problems that can be understood under the proposed settings.
>
>
> $\textbf{Regarding other related works.}$ We thank the reviewer for bringing those related works [a-d] into our attention. In what follows, we briefly compare them with our paper, and we are happy to have the discussions included in the updated manuscript. Our paper attempts to solve undetermined stochastic combinatorial problems using input-solution pairs without knowing the true configuration distribution, while the focuses and methodologies in [a-d] are somehow different from ours:
> -	[a] presents an interesting framework that solves hard combinatorial problems by aligning the solutions of the target problem with the solutions to an easy problem; in contrast, our paper seeks the alignment between inputs and their optimal solutions.
> -	[b] focuses on computing the gradient of the learning framework composed of neural networks and combinatorial processes. This is indeed a promising method for allowing USCO-Solver to handle extra features associated with the inputs or solutions. In our paper, we only consider the input and solutions that are pure combinatorial objects.
> -	[c] studies the problem of approximating the instance of two-stage stochastic integer programming using representative scenarios, which is conceptually similar to [a] in that they both attempt to connect one optimization problem with another one. Different from our paper, a) the training phase in [c] requires to know the original instance, and b) [c] focuses on approximating the instance parameters (analogous to the true distributions under our context) rather than the optimization effect.
> -	[d] designs a reinforcement learning method to solve the two-stage stochastic integer programming, where the objective function is given approximately by iid samples.
>
> $\textbf{[a]}$ [Parmentier, A. (2021). Learning to approximate industrial problems by operations research classic problems. Operations Research.]
>
> $\textbf{[b]}$ [Pogančić, Marin Vlastelica, et al. "Differentiation of blackbox combinatorial solvers." International Conference on Learning Representations. 2019.]
>
> $\textbf{[c]}$ [Bengio, Yoshua, et al. "A learning-based algorithm to quickly compute good primal solutions for Stochastic Integer Programs." International Conference on Integration of Constraint Programming, Artificial Intelligence, and Operations Research. Springer, Cham, 2020.]
>
> $\textbf{[d]}$ [Nair, Vinod, et al. "Learning Fast Optimizers for Contextual Stochastic Integer Programs." UAI. 2018.]
>
> $\textbf{Training complexity.}$ We adopt the standard one-slack cutting-plane algorithm (Algorithm 3 in [e]) for training, which involves a hyperparameter $\epsilon$ (in addition to the hyperparameters $C$ and $\eta$ in our formulation) to control the stopping criteria. Following directly from Corollary 2 in [e], for fixed $C, \eta$ and $\epsilon$, the time complexity is $O(m*A+m^2+mK+K^3)$ where $m$ is the number of training instances, $A$ is the running time of the oracle (in Sec 2.1), and $K$ is the number of sampled configurations. We adopt the implementation provided by [f], where we have $C=0.0001$, $\eta=1$, and $\epsilon=0.001$. We believe that training complexity is an important issue, and we will add such discussions to the updated version.
>
> $\textbf{[e]}$ Joachims, Thorsten, Thomas Finley, and Chun-Nam John Yu. "Cutting-plane training of structural SVMs." Machine learning 77, no. 1 (2009): 27-59.
>
> $\textbf{[f]}$ Müller, Andreas C., and Sven Behnke. "PyStruct: learning structured prediction in python." J. Mach. Learn. Res. 15, no. 1 (2014): 2055-2060.
>
> $\textbf{Line 188.}$ We follow the convention that low-order terms are often ignored when formulating empirical risk minimization by minimizing learning bounds (e.g., [g, h]), which is also due to the purpose of easing the optimization burden.
>
> $\textbf{[g]}$ McAllester, David. "Generalization bounds and consistency." Predicting structured data (2007): 247-261.
>
> $\textbf{[h]}$ Wu, Yuanbin, et al. "A learning error analysis for structured prediction with approximate inference." Proceedings of the 31st International Conference on Neural Information Processing Systems. 2017.
>
> $\textbf{Line 220.}$ We mean that the result of one problem instance is obtained with five repetitions, each of which contains randomly selected data. For example, in the SSP problem, 160 training pairs and 6400 testing pairs are randomly selected in each run, and the experiment is repeated five times based on which we compute the average performance. We will clarify this in the updated version.

---

### Decision · Program_Chairs · 2021-09-27

**Decision:**

Accept (Poster)

**Comment:**

The paper considers learning to solve the stochastic combinatorial optimization problem, where the underlying distribution over configuration space is unknown---a problem class which the authors refer to as *undetermined* stochastic combinational problem.

Assuming that a deterministic (approximation) oracle and the corresponding input-solutions pairs are given as training data, the authors propose to use combinatorial kernels (with random configurations) to train a large-margin machine. There was initial disagreement among the reviewers in the applicability of the proposed algorithms beyond the combinatorial optimization problems mentioned in the paper; however, all reviewers find the proposed approach interesting and the theoretical analysis of training and generalization bounds (which is applied to the approximation ratio of the learned solution) insightful.

The authors are encouraged to address the concerns raised in the reviews; in particular, to clarify the missing references, algorithmic details, computation complexity, and applicability to realistic domains.